# WorldAlignment: Benchmarking Expert-Level Human Preference Alignment across Domains and Aspects

## Abstract

Large Language Models (LLMs) demonstrate remarkable capabilities, yet aligning their behavior with human preferences remains both challenging and essential. Human evaluation of model outputs is often costly and must account for diverse user preferences. To address this, recent methods leverage LLM-as-a-judge to assess alignment quality, which achieves higher agreement with human judgments while being more cost-effective. However, existing widely used benchmarks such as AlpacaEval 2.0 primarily rely on simplistic, instruction-following data pairs derived from general user preferences. These benchmarks are insufficient for evaluating complex, domain-specific, and nuanced scenarios, and many are outdated for current alignment evaluation. To overcome these limitations, we introduce WorldAlignment, an expert-level, multi-domain human preference benchmark designed for efficient and comprehensive evaluation of alignment capabilities. WorldAlignment provides more challenging, higher-quality, and diverse preference pairs across multiple domains, enabling more robust alignment assessment. Our evaluation results show that WorldAlignment offers comprehensive insights into both SoTa and post-trained models, establishing a modern benchmark for domain-oriented alignment. Furthermore, WorldAlignment supports evaluation across various dimensions, including instruction-following, mathematical reasoning, and code-related tasks, providing a holistic view of alignment performance. Through our evaluation, we find that several state-of-the-art alignment-tuned models still exhibit substantial performance gaps compared to GPT-4-level models on our benchmark, highlighting critical limitations and directions for future improvement. Our code and data will be available at https://anonymous.4open.science/r/WorldAlignment.

## 1 Introduction

The advancement of large language models (LLMs) necessitates robust evaluation frameworks that can assess model performance across diverse, real-world scenarios. While evaluation in constrained domains with objective metrics remains relatively straightforward (Novikova et al., 2017; Yeh et al., 2021), the assessment of open-ended language generation presents significant challenges, particularly when models must demonstrate competency across multiple specialized knowledge domains.

Evaluation paradigms, including reference-based metrics such as BERTScore (Zhang* et al., 2020), face fundamental limitations when applied to complex, multi-domain scenarios. In particular, constructing comprehensive reference sets that fully capture the range of valid responses across specialized fields—such as mathematical reasoning or code generation—is highly challenging, making these approaches inadequate for evaluating modern LLMs.

To address these limitations, the research community has increasingly adopted reference-free evaluation methodologies that leverage high-capability LLMs as judges. While approaches such as AlpacaEval (Li et al., 2023), MT-Bench (Zheng et al., 2023), and WildBench (Lin et al., 2024) have demonstrated promising correlations with human preferences, they are often susceptible to spurious correlations, including response length bias, formatting preferences, and positional effects (Li et al., 2023; Zheng et al., 2023; Koo et al., 2023; Wang et al., 2023; Wu & Aji, 2023).

Figure 1: Overview of the WorldAlignment designed for comprehensive LLM alignment evaluation.

Recent efforts such as AlpacaEval 2.0 (Dubois et al., 2025) have attempted to address these biases through regression-based debiasing strategies, treating spurious correlates like response length as undesirable mediators in causal inference frameworks (VanderWeele, 2015; Hernán & Robins, 2010). While AlpacaEval 2.0 demonstrates improved correlation with human preferences compared to its predecessor and shows greater robustness to simple gaming strategies, it maintains the fundamental limitation of focusing primarily on instruction-following tasks.

More critically, existing benchmarks suffer from limited domain coverage, concentrating predominantly on conversational and general-purpose tasks while neglecting the specialized knowledge domains that are increasingly central to real-world LLM applications. This gap becomes particularly pronounced when evaluating models' ability to navigate complex scenarios requiring domain expertise, such as advanced mathematical problem-solving, sophisticated code generation, or nuanced reasoning across multiple knowledge areas simultaneously.

As shown in Figure 1, to bridge this evaluation gap we introduce **WorldAlignment**, a multi-aspect human preference benchmark designed to assess LLM alignment across three critical dimensions: instruction following, mathematical reasoning, and code-related tasks. Unlike conventional benchmarks such as AlpacaEval 2.0, which primarily emphasize surface-level instruction adherence, WorldAlignment incorporates complex preference pairs that demand deep domain understanding and the integration of specialized knowledge, enabling a more rigorous and nuanced assessment of model capabilities in realistic scenario.

Our contributions are summarized as follows:

- We present WORLDALIGNMENT, to our knowledge the first comprehensive, multi-aspect evaluation benchmark that goes beyond conventional instruction-following tasks by incorporating mathematical reasoning and code-related preference alignment, enabling fine-grained assessment across specialized domains.

- We provide empirical evidence that WorldAlignment's increased complexity and domain-oriented design supports more rigorous evaluation of alignment, particularly for expert-level tasks. We primarily evaluate the benchmark using state-of-the-art LLMs to establish a standard reference for human preference alignment research.

- We show that many academic post-training and alignment-tuned models still lag behind GPT-4-level performance. While these models often optimize for instruction-following, they exhibit

substantial gaps in math and code alignment with human preferences, highlighting key limitations and directions for future improvement.

## 2 BACKGROUND AND RELATED WORK

**Reference-free evaluation metrics** Reference-free evaluation metrics, which assess model performance on open-ended tasks without relying on reference answers, have evolved from classical approaches (Louis & Nenkova, 2013) to sophisticated neural methods (Kryscinski et al., 2020; Sinha et al., 2020; Goyal & Durrett, 2020). While recent neural approaches achieve inter-annotator agreement levels comparable to human evaluation, they remain susceptible to spurious correlations including perplexity and response length (Durmus et al., 2022). The emergence of LLM-based evaluation frameworks has addressed many of these limitations through zero-shot, reference-free assessment (Zheng et al., 2023; Dubois et al., 2023; Li et al., 2023; Lin et al., 2024), establishing new paradigms for automated model evaluation.

**Chatbot Arena and Human Preference Collection** Chatbot Arena (Zheng et al., 2023) represents the gold standard for human preference evaluation in conversational AI. The platform enables real-world human evaluation through live interactions, where users simultaneously query pairs of anonymized models and provide comparative quality assessments. These head-to-head comparisons are subsequently converted to Elo ratings (Elo, 1978), providing robust model rankings. While Chatbot Arena offers highly desirable properties—including real user engagement and dynamic instruction diversity that resists benchmark saturation—its reliance on extensive human evaluation renders it prohibitive for iterative model development due to cost and latency constraints.

**Automated Evaluation Benchmarks** Recent automated evaluation frameworks have sought to approximate human preferences while maintaining computational efficiency. MT-Bench (Zheng et al., 2023) provides multi-turn conversation evaluation, while WildBench (Lin et al., 2024) focuses on diverse, real-world scenarios. AlpacaEval 2.0 (Dubois et al., 2025) represents a significant advancement in this space, incorporating length-controlled win-rates that achieve a Spearman correlation of 0.98 with Chatbot Arena while requiring less than \$10 in computational costs and completing evaluation in under 3 minutes. This framework contains extensive pre-computed LLM-based pairwise comparisons across over 120 models and 805 instructions, making it widely adopted by the research community.

## 3 WORLDALIGNMENT

### 3.1 PROBLEM FORMULATION

Following established evaluation paradigms, we define multi-domain pairwise evaluation as follows. Given an instruction $x$ sampled from a domain-specific distribution $p(x)$, where the domain $d \in \{\text{instruction-following}, \text{mathematical reasoning}, \text{code-related}\}$, a baseline model generates response $z_b$ and the evaluated model generates response $z_m$. A human annotator produces preference $y \in \{0, 1\}$ indicating whether the candidate response $z_m$ ($y = 1$) is superior to the baseline response $z_b$ ($y = 0$) according to domain-specific quality criteria. An automated evaluation framework such as WorldAlignment serves as predictor $f(z_m, z_b, x, d)$ that approximates human preference $p(y|z_m, z_b, x, d)$ while incorporating domain-specific evaluation criteria. The domain-specific win rate is computed as $\text{win rate}_d(m, b) = 100 \cdot \mathbb{E}_{x \sim p_d(x)}[f(z_m, z_b, x, d)]$, where $p_d(x)$ represents the instruction distribution for domain $d$.

### 3.2 DATASET COMPOSITION

Inspired by the theory of world knowledge compression and recovery in persona-based models (Ge et al., 2025), we construct the WORLDALIGNMENT benchmark entirely from high-quality synthetic data. The use of personas is central to our design: by conditioning data generation on diverse personas, we can systematically control the style and difficulty of the generated prompts. This brings two key benefits. First, more complex personas encourage the model to generate richer and more challenging instructions, leading to broader coverage of task types. Second, persona-guided

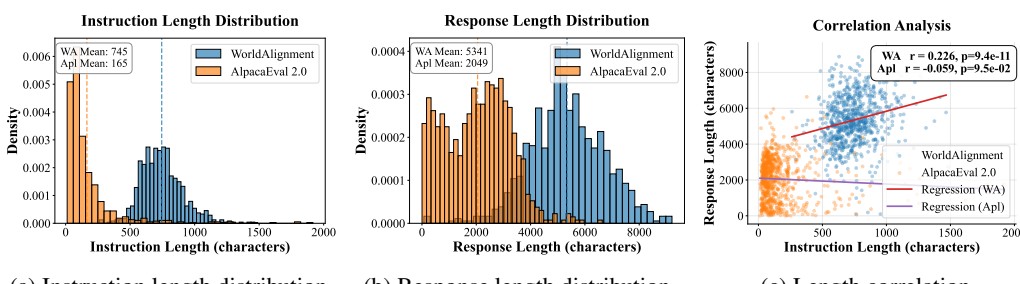

(a) Instruction length distribution.    (b) Response length distribution.    (c) Length correlation.

Figure 2: Length distribution and correlation analysis comparing **WorldAlignment (WA)** and **AlpacaEval 2.0 (Apl)**. (a) WA instructions are substantially longer and more diverse than Apl. (b) WA responses are correspondingly longer and more elaborated. (c) WA shows a strong positive correlation between instruction and response length, while Apl exhibits only a weak relationship.

generation reduces reliance on few-shot exemplars that dominate current data synthesis pipelines, thereby mitigating both data contamination and few-shot bias. Concretely, we collect a set of domain personas $\{p_i\}_{i=1}^{N}$, each used exactly once to generate prompts across three aspects: instruction following, mathematical reasoning, and code-related tasks. Using GPT-4o as the generator $G$, for each persona $p_i$ we obtain prompt–response pairs:

$$(x_i^d, y_i^d) = G(p_i, d), \quad d \in \{\text{inst}, \text{math}, \text{code}\}. \tag{1}$$

We then filter and clean the dataset to ensure quality, removing samples that are harmful, biased, or offensive in each aspect. For evaluation, we retain 800 high-quality examples per aspect. In addition, we provide detailed persona-guided templates and representative examples in Appendix C.

### 3.2.1 LENGTH DISTRIBUTION ANALYSIS

To provide a fair comparison with AlpacaEval 2.0, we analyze both instruction and response length distributions in WorldAlignment (inst). As shown in Figure 2a, WorldAlignment instructions exhibit a substantially broader length distribution, extending to longer and more complex prompts. This reflects its design toward domain-specific tasks that require detailed context, multiple constraints, or elaborate problem statements beyond simple question–answer formats. As shown in Figure 2b, the contrast is even sharper in response lengths. WorldAlignment responses typically range from 4,000 to 8,000 characters, whereas AlpacaEval 2.0 responses are mostly concentrated between 2,000 and 3,000 characters. This indicates that WorldAlignment emphasizes comprehensive, elaborated answers that demand depth of understanding and sustained engagement with the prompt.

To further analyze the correlation between instruction and response lengths, Figure 2c presents their relationship across datasets. In WorldAlignment, instruction length shows a significant positive correlation with response length ($r = 0.226$, $p = 9.4 \times 10^{-11}$), suggesting that longer prompts naturally elicit more detailed responses. By contrast, AlpacaEval 2.0 exhibits only a weak and statistically insignificant correlation ($r = -0.059$, $p = 9.5 \times 10^{-2}$), consistent with its narrower range of prompt complexity. These results demonstrate that WorldAlignment captures richer prompt–response dynamics, making it a more demanding benchmark for evaluating long-form generation.

### 3.2.2 TASK DIFFICULTY, FEASIBILITY AND QUALITY ASSESSMENT

To ensure consistent evaluation standards, we assessed each instruction–response pair along three dimensions using GPT-4o guided by well-defined criteria. **Difficulty** reflects task complexity, required domain knowledge, cognitive load, and technical sophistication, ranging from elementary knowledge (1–2) to cutting-edge research challenges (9–10). **Feasibility** measures whether instructions describe realistic and executable tasks, judged by contextual reasonableness, logical consistency, and real-world applicability (1–2: unrealistic; 9–10: highly practical). **Quality** evaluates the response itself in terms of accuracy, completeness, clarity, alignment, and educational value (1–2: poor; 9–10: excellent). Further details are provided in Appendix D, and Figure 3 reports the resulting distributions.

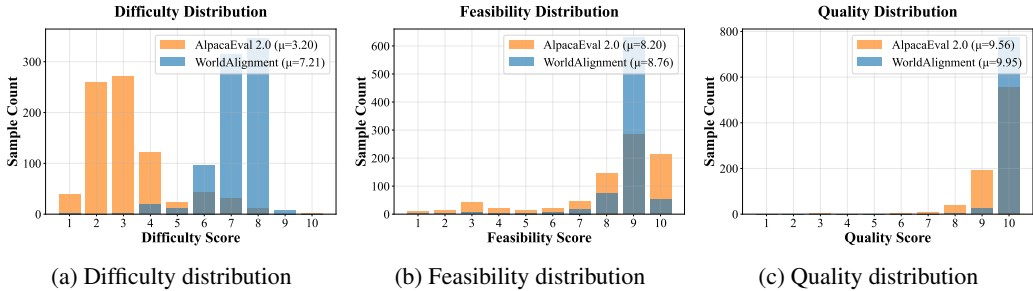

(a) Difficulty distribution     (b) Feasibility distribution     (c) Quality distribution

Figure 3: Comparison of distribution across three evaluation dimensions between **WorldAlignment (WA)** and **AlpacaEval 2.0 (Apl)**. (a) Difficulty: WA tasks span a broader and higher range of difficulty than Apl. (b) Feasibility: WA shows higher overall feasibility, while Apl includes more low-feasibility cases. (c) Quality: WA responses demonstrate consistently higher quality levels.

On difficulty, AlpacaEval 2.0 centers around an average score of $\mu = 3.20$, reflecting predominantly intermediate-level tasks. By contrast, WorldAlignment shifts toward expert-level tasks, with $\mu = 7.21$, requiring deeper domain-specific knowledge and advanced reasoning. Regarding feasibility, WorldAlignment maintains high implementability of its tasks, with an average score of $\mu = 8.76$, compared to $\mu = 8.20$ in AlpacaEval 2.0. This ensures that tasks are both challenging and practically realistic. In terms of quality, WorldAlignment achieves an average score of $\mu = 9.95$, slightly higher than AlpacaEval 2.0 ($\mu = 9.56$). Most instruction–response pairs score between 9 and 10, reflecting rigorous construction standards and enabling preference-based evaluations on high-quality content.

### 3.3 MULTI-DOMAIN, MULTI-ASPECT EVALUATION FRAMEWORK

Inspired by the length-controlled evaluation methodology of AlpacaEval 2.0, WorldAlignment ensures a robust preference assessment across diverse domains. Unlike AlpacaEval 2.0, which focuses primarily on general instruction-following, our framework encompasses three critical aspects for real-world applications: instruction following, mathematical reasoning, and code generation.

We maintain a regression-based approach to control for spurious correlations, particularly length bias, while extending the scope to capture domain-specific preference patterns. Unlike previous works that primarily addressed length bias, we propose a novel multi-domain regression framework that models preference outcomes across different task categories. Specifically, a logistic regression model is fitted to predict preference outcomes, and length-corrected win rates are obtained by setting the length term to zero, ensuring fair comparison between models with varying verbosity.

#### 3.3.1 MULTI-DOMAIN REGRESSION MODEL

Building on the AlpacaEval 2.0 methodology, we employ a logistic regression to model WorldAlignment predictions, which extends the original framework to accommodate domain heterogeneity. This approach maintains the three core terms—model, length, and prompt—while incorporating a more nuanced, domain-aware analysis. The model is defined as follows:

$$q_{\theta,\phi,\psi}(y=1|z_m, z_b, m, b, x, d) := \text{logistic}\Big(\underbrace{\theta_m - \theta_b}_{\text{Model}} + \underbrace{\phi_{m,b} \cdot \tanh\Big(\frac{\text{len}(z_m) - \text{len}(z_b)}{\text{std}(\text{len}(z_m) - \text{len}(z_b))}\Big)}_{\text{Length}} + \underbrace{d((\psi_m - \psi_b)\gamma_x)}_{\text{Prompt}}\Big)$$

(2)

where $d$ denotes the domain category. The model and prompt terms are consistent with the original AlpacaEval 2.0 framework. They capture the log-linear contribution of the model pair $(m, b)$ and the intrinsic difficulty ($\gamma$) of each prompt to the baseline win rate. The length term remains linear in a normalized length feature, which is standardized to unit variance and then transformed using the $\tanh$ function to account for the diminishing returns on log odds.

This formulation preserves the essential identity and symmetry properties of the original framework: $q(y = 1 \mid z_b, z_b, b, b, x, d) = 0.5$ and $q(y = 1 \mid z_m, z_b, m, b, x, d) = 1.0 - q(y = 1 \mid z_b, z_m, b, m, x, d)$, ensuring mathematical consistency across domain-specific evaluations.

**AlpacaEval 2.0**

**Instruction 1:** *"What are the names of some famous actors that started their careers on Broadway?"*

**Difficulty Score: 2**

**Instruction 2:** *"I like to host guests at my home from time to time, and I am gathering recipes of different dishes and drinks to keep things interesting. I am interested in trying some Georgian dishes. Can you give me a recipe for Kubdari?"*

**Difficulty Score: 3**

**WorldAlignment**

**Instruction 1:** *"Provide a detailed guide on integrating GPRS technology into an embedded system, including selecting suitable modules, implementing a stable and efficient software stack, and ensuring reliable data transmission. The solution should be easy to use, require minimal technical expertise, and be compatible with other network protocols. Additionally, discuss best practices for optimizing performance and ensuring seamless integration with existing systems."*

**Difficulty Score: 7**
**Domain:** Computer Science

**Instruction 2:** *"Provide an analysis of the Russian economy in response to international sanctions and oil price fluctuations, focusing on the effects on macroeconomic indicators such as GDP growth, inflation, and exchange rates. Additionally, discuss the impact of the agricultural and food import ban on consumer prices."*

**Difficulty Score: 8**
**Domain:** General

Figure 4: Instruction Comparison of WorldAlignment and AlpacaEval 2.0 benchmarks. WorldAlignment presents substantially higher task complexity compared to AlpacaEval 2.0. While AlpacaEval 2.0 mainly consists of straightforward factual queries that can be addressed with short list-style responses, WorldAlignment emphasizes in-depth analyses requiring contextual reasoning, critical evaluation, and integration of specialized domain knowledge.

### 3.3.2 LENGTH-CORRECTED MULTI-DOMAIN WIN RATES

Using the model from Equation 2, we obtain the length-corrected win rate estimate by setting the length difference to zero:

$$\text{winrate}^{LC}(m, b, d) = 100 \cdot \mathbb{E}_x[\text{logistic}(\theta_m - \theta_b + (\psi_m - \psi_b)\gamma_x)] \qquad (3)$$

This approach facilitates the computation of both overall performance metrics and granular, domain-specific assessments. This capability allows our framework to reveal performance disparities across instruction following, mathematical reasoning, and code generation.

## 4 RESULTS

### 4.1 EVALUATION SETTING

To establish a robust evaluation framework for modern large language models, we conduct comprehensive assessments across state-of-the-art models spanning both open-source and proprietary architectures. We utilize GPT-4o responses as our baseline reference, given its widespread community acceptance as an advanced and human-aligned model. Our evaluation employs a dual-judge system: GPT-4o serves as the primary evaluator, while GPT-4.1-Mini provides complementary assessment

Table 1: WorldAlignment evaluation results across three core aspects. Models are judged by GPT-4o and GPT-4.1-Mini using two metrics: **Win Rate (WR)** and **Length-Controlled Win Rate (LC)**, which measure the proportion of responses preferred over the GPT-4o baseline under the same prompt. The best performance in each category is highlighted in bold.

| Model | GPT-4o | | GPT-4.1-Mini | | Avg Length |
|---|---|---|---|---|---|
| | LC (%) | WR (%) | LC (%) | WR (%) | |
| **Instruction Following** | | | | | |
| GPT5 | 46.49 | **68.34** | 49.91 | 72.83 | 7145 |
| GPT-4.1-2025-04-14 | **54.41** | 60.46 | **64.09** | 71.34 | 6023 |
| GPT-4.1-Mini-2025-04-14 | 44.53 | 53.37 | 52.08 | 66.13 | 6441 |
| O1-2024-12-17 | 33.11 | 55.35 | 40.03 | 69.61 | 8318 |
| O3-Mini-2025-01-31 | 36.01 | 52.04 | 37.72 | 59.62 | 7353 |
| GPT-4o-Mini-2024-07-18 | 38.85 | 34.73 | 21.08 | 19.60 | 4956 |
| Gemma-3-27B-IT | 29.75 | 59.28 | 42.37 | **76.21** | 8533 |
| **Mathematical Reasoning** | | | | | |
| GPT5 | **65.09** | **80.28** | **78.51** | **90.10** | 5697 |
| GPT-4.1-2025-04-14 | 60.84 | 72.63 | 76.44 | 87.15 | 5654 |
| GPT-4.1-Mini-2025-04-14 | 54.52 | 65.49 | 73.19 | 86.09 | 6108 |
| O1-2024-12-17 | 39.45 | 61.08 | 57.27 | 78.82 | 4679 |
| O3-Mini-2025-01-31 | 53.31 | 73.70 | 74.44 | 86.91 | 4603 |
| GPT-4o-Mini-2024-07-18 | 21.40 | 29.64 | 15.07 | 24.03 | 3343 |
| Gemma-3-27B-IT | 26.67 | 40.33 | 38.68 | 52.62 | 4683 |
| **Code Generation** | | | | | |
| GPT5 | 44.07 | 43.62 | 54.01 | 52.73 | 4790 |
| GPT-4.1-2025-04-14 | **47.37** | **50.10** | **70.30** | 71.89 | 5508 |
| GPT-4.1-Mini-2025-04-14 | 43.12 | 48.22 | 69.83 | **74.94** | 6235 |
| O3-Mini-2025-01-31 | 31.09 | 39.66 | 52.43 | 61.91 | 5910 |
| O1-2024-12-17 | 22.83 | 32.08 | 40.23 | 52.15 | 6365 |
| GPT-4o-Mini-2024-07-18 | 14.23 | 9.64 | 17.69 | 12.81 | 4013 |
| Gemma-3-27B-IT | 12.51 | 16.66 | 27.21 | 31.03 | 4766 |

as a cost-effective secondary judge. The evaluation framework encompasses three fundamental capabilities: instruction following, mathematical reasoning, and code generation. We employ two critical metrics: Win Rate (WR), which measures the raw proportion of responses preferred over the baseline, and Length-Controlled Win Rate (LC), which mitigates verbosity bias by normalizing for response length differences. This dual-metric approach reveals performance patterns that conventional benchmarks often obscure, providing more nuanced insights into true model capabilities beyond superficial response quality indicators.

## 4.2 Performance Analysis Across Advanced Language Models

**Instruction Following:** GPT-4.1-2025-04-14 demonstrates superior length-controlled performance with LC scores of 54.41% and 64.09% under GPT-4o and GPT-4.1-Mini evaluation respectively, indicating robust instruction adherence independent of response length. However, GPT5 achieves the highest raw win rate (68.34% under GPT-4o evaluation) despite lower LC performance (46.49%), suggesting that longer responses may bias evaluators toward favorable judgments. Notably, Gemma-3-27B-IT exhibits exceptional performance under GPT-4.1-Mini evaluation (76.21% WR) while maintaining reasonable LC scores (42.37%), indicating effective instruction following despite being an open-source model. The substantial gap between WR and LC metrics across most models (averaging 15-20 percentage points) underscores the critical importance of length-controlled evaluation.

**Mathematical Reasoning:** GPT5 demonstrates overwhelming superiority in mathematical reasoning, achieving the highest performance across all metrics with LC scores of 65.09% and 78.51%

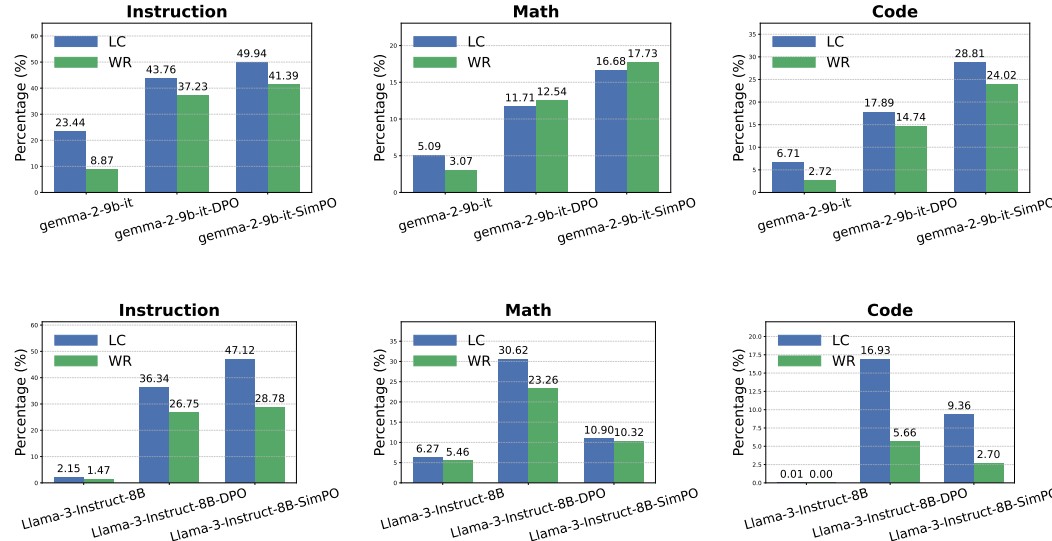

Figure 5: Performance comparison on Instruction, Math, and Code tasks for two mainstream post-training model series. We observe that SimPO generally outperforms DPO across most settings. But for the Llama series, SimPO shows weaker performance than DPO on mathematical reasoning and code generation tasks, highlighting architecture-specific differences in optimization effectiveness.

under both evaluators, and exceptional WR scores reaching 90.10% under GPT-4.1-Mini evaluation. GPT-4.1 series models maintain competitive performance, with GPT-4.1-2025-04-14 achieving 60.84% and 76.44% LC scores respectively. The performance gap between frontier models and smaller alternatives is particularly pronounced in this domain, with GPT-4o-Mini exhibiting significantly lower performance (21.40% LC under GPT-4o evaluation) and Gemma-3-27B-IT achieving only 26.67% LC performance. This suggests that mathematical reasoning requires substantial model capacity and sophisticated training procedures.

**Code Generation:** The code generation domain exhibits more balanced performance across models, with GPT-4.1-2025-04-14 leading in LC performance (47.37% under GPT-4o, 70.30% under GPT-4.1-Mini) and demonstrating consistent superiority in length-controlled metrics. Interestingly, GPT-4.1-Mini-2025-04-14 achieves the highest WR under GPT-4.1-Mini evaluation (74.94%), suggesting effective code generation capabilities in the mini model variant. The substantial performance difference between evaluators (with GPT-4.1-Mini consistently rating models higher) indicates potential evaluator-specific biases in code quality assessment. Notably, all models exhibit relatively modest performance in this domain compared to mathematical reasoning, with even the best-performing models achieving LC scores below 50% under GPT-4o evaluation.

### 4.3 IMPACT OF POST-TRAINING METHODS ON MODEL PERFORMANCE

Figure 5 illustrates the effectiveness of two different post-training approaches: Direct Preference Optimization (DPO) (Rafailov et al., 2024) and Simple Preference Optimization (SimPO) (Meng et al., 2024) across the Gemma and Llama model families. The analysis reveals distinct optimization patterns between model architectures and training methodologies, with notable differences between base models and their optimized variants.

**Gemma Series Analysis.** For *instruction following*, the base Gemma-2-9b-it achieves only 23.44% LC and 8.87% WR. DPO substantially improves to 43.76% LC / 37.23% WR, while SimPO yields the best results with 49.94% LC / 41.39% WR, highlighting its advantage in aligning models with human preferences. In *math reasoning*, the base model starts from 5.09% LC / 3.07% WR. DPO improves to 11.71% LC / 12.54% WR, and SimPO further advances to 16.68% LC / 17.73% WR, showing that WR reveals more consistent improvements even when LC alone underestimates DPO's effect. For *code generation*, Gemma-2-9b-it performs poorly (6.71% LC / 2.72% WR). DPO raises

Table 2: Evaluation on the top five domains from the WorldAlignment(inst) benchmark: general knowledge, medicine, biology, history, and engineering. Metrics reported are LC, WR, Avg Len, and sample size (N), with best results highlighted in bold.

| Domain | GPT-4.1-Mini-2025-04-14 | | | | GPT-4o-Mini-2024-07-18 | | | | O3-Mini-2025-01-31 | | | |
|---|---|---|---|---|---|---|---|---|---|---|---|---|
| | LC | WR | Avg Len | N | LC | WR | Avg Len | N | LC | WR | Avg Len | N |
| General | **34.95** | **50.13** | 6360 | 145 | 33.35 | 30.86 | 4894 | 145 | 24.30 | 49.42 | 7428 | 145 |
| Medicine | **45.16** | **52.39** | 6487 | 64 | 34.77 | 24.24 | 4857 | 64 | 26.57 | 49.86 | 7449 | 64 |
| Biology | 26.50 | 47.59 | 6244 | 53 | 36.91 | 29.21 | 4741 | 53 | 23.09 | **50.70** | 7199 | 53 |
| History | 35.01 | **58.00** | 6887 | 50 | **44.93** | 42.38 | 5259 | 50 | 32.38 | 58.20 | 7566 | 50 |
| Engineering | 35.93 | **59.27** | 6598 | 27 | **42.04** | 45.95 | 5090 | 27 | 29.04 | 56.44 | 7515 | 27 |

performance to 17.89% LC / 14.74% WR, and SimPO achieves the highest 28.81% LC / 24.02% WR, underscoring the effectiveness of SimPO for structured tasks.

**Llama-3-Instruct Series Analysis.** Unlike Gemma, the base Llama model exhibits higher LC scores but weak WR due to verbosity. In *instruction following*, the base model records 2.15% LC / 1.47% WR. DPO improves to 36.34% LC / 26.75% WR, while SimPO excels at 47.12% LC / 28.78% WR, confirming SimPO's superior robustness. For *math reasoning*, the base model achieves 6.27% LC / 5.46% WR. DPO reaches 30.62% LC / 23.26% WR, whereas SimPO underperforms with only 10.90% LC / 10.32% WR—indicating that SimPO is not universally better in mathematical reasoning. In *code generation*, the base Llama almost fails (0.01% LC / 0.00% WR). DPO improves moderately to 16.93% LC / 5.66% WR, while SimPO lags behind with 9.36% LC / 2.70% WR, showing that preference optimization benefits coding but SimPO may not consistently outperform DPO for this architecture. Future work may further investigate this interesting phenomenon.

### 4.4 DOMAIN-SPECIFIC PERFORMANCE ANALYSIS

To further evaluate domain-level performance, we select the top five domains from the WorldAlignment benchmark for detailed analysis, as reported in Table 2. Results highlight distinct strengths among the three mini-model series. GPT-4.1-Mini-2025-04-14 delivers the most consistent LC performance across domains, with notable strength in medicine (45.16% LC / 52.39% WR) and general knowledge (34.95% LC / 50.13% WR). O3-Mini-2025-01-31 excels mainly in WR, achieving leading results in biology (50.70%) and history (58.20%) but at the cost of lower LC. GPT-4o-Mini-2024-07-18 shows overall weaker results, though it remains competitive in history (44.93% LC) and engineering (42.04%), suggesting potential domain-specific optimization benefits.

Medical and engineering tasks illustrate these contrasts well. GPT-4.1-Mini not only achieves the highest LC but also maintains solid WR, indicating concise yet accurate reasoning. O3-Mini, despite generating long responses, struggles to convert verbosity into higher LC, while GPT-4o-Mini offers moderate but stable results. These patterns suggest that domain complexity interacts differently with each optimization approach, shaping the trade-off between conciseness and coverage.

Response length provides further insight. O3-Mini produces the longest outputs (7k–7.5k tokens), which correlate with high WR but low LC, revealing a systematic length bias. By comparison, GPT-4.1-Mini balances brevity and accuracy more effectively, achieving competitive WR while preserving superior LC. This underscores the importance of length-controlled metrics to ensure fair cross-domain evaluation, as raw WR alone may overstate performance for verbose models.

## 5 CONCLUSIONS AND DISCUSSIONS

In this paper, we introduce WorldAlignment, a comprehensive benchmark for expert-level human preference alignment across domains and tasks, including instruction-following, mathematical reasoning, and code generation, all requiring expert-level understanding. Evaluation across state-of-the-art models reveals substantial performance variation, highlighting specialized strengths and weaknesses often obscured by general-purpose benchmarks. Notably, even alignment-tuned models fall short of GPT-4-level performance, exposing persistent gaps and opportunities for improvement. WorldAlignment thus provides a scalable and rigorous framework for fine-grained analysis of LLM capabilities in real-world, expertise-driven scenarios.

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

## A  LIMITATIONS AND FUTURE WORK.

While our work focuses on evaluating LLM alignment across three core aspects and five specialized domains, the framework could be extended to additional domains such as scientific research, legal reasoning, and creative writing. Expanding WorldAlignment to incorporate multilingual evaluation and cross-cultural preference alignment represents another promising direction. Additionally, integrating dynamic difficulty adjustment and adaptive evaluation strategies could further enhance the framework's ability to provide fine-grained assessment across varying complexity levels within each domain.

## B  ADDITIONAL EXPERIMENT DETAILS

### B.1  DOMAIN DISTRIBUTION AND SPECIALIZATION

As illustrated in Figures 6a and 6b, the dataset composition reveals distinct patterns across our evaluation framework. The general domain distribution (Figure 6a) shows a balanced representation of core LLM capabilities, while the specialized domain analysis (Figure 6b) demonstrates targeted coverage of critical knowledge areas.

The specialized domain analysis reveals that while the majority of our dataset comprises general-purpose tasks (145 instances labeled as "General"), the remaining portion provides substantial coverage across critical specialized fields. Statistical mathematics leads the specialized domains with 11 instances, followed by marine biology (8 instances), and several other fields including microbiology, conservation biology, dermatology, and paleontology (3-5 instances each). This distribution ensures comprehensive evaluation across both general capabilities and domain-specific expertise, addressing the growing need for specialized AI assistance in professional and academic contexts.

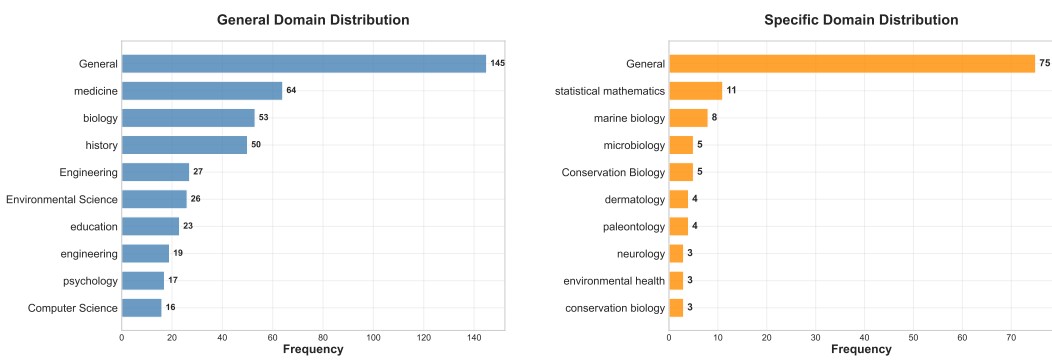

(a) Top 10 general domains in WorldAlignment     (b) Top 10 specialized domains in WorldAlignment

Figure 6: Domain composition analysis showing both general and specialized categories in WorldAlignment. In our benchmark, we focus exclusively on the General domain.

## C PERSONA-GUIDED PROMPT TEMPLATES

We provide the detailed templates used to generate instruction, math, and code prompts conditioned on different personas.

---

**Instruction Prompt Template**

Guess a prompt that the following persona may ask you to do:{persona}

**Note:**

1. The prompt should be informative and specific.
2. Your output should start with "User prompt:".

---

**Math Prompt Template**

Create a math problem related to the following persona: {persona}

**Note:**

1. The math problem should be challenging and involve advanced mathematical skills and knowledge. Only top talents can solve it correctly.
2. You should make full use of the persona description to create the math problem to ensure that it is unique and specific to the persona.
3. Your response should always start with "Math problem:". Do not include a solution to the created problem.
4. The math problem should include no more than two sub-problems.

---

**Code Prompt Template**

Create a coding problem or programming task related to the following persona: {persona}

**Note:**

1. The coding problem should be challenging and require solid programming skills. It should test both algorithmic thinking and implementation abilities.
2. You should make full use of the persona description to create a coding problem that is unique and specific to the persona's background, interests, or profession.

---

3. Your response should always start with "Code problem:". Do not include the solution or implementation code.

4. The problem should be well-defined with clear input/output requirements and constraints.

5. Consider the persona's expertise level and create an appropriately challenging problem relevant to their field or interests.

6. The problem can involve any programming language or technology stack that fits the persona's background.

# D   EVALUATION PROMPT

## D.1   DIFFICULTY EVALUATION PROMPT

We employ the following system prompt to evaluate the difficulty level of instructions:

**System Prompt:** You are an expert evaluator for assessing the difficulty level of instructions/questions. Please evaluate the given instruction on a scale of 1-10 based on:

- Complexity of the task or question
- Level of domain knowledge required
- Cognitive load needed to understand and process
- Technical depth and sophistication

Scoring guidelines:

- 1-2: Very basic, elementary level knowledge
- 3-4: Intermediate, requires some specialized knowledge
- 5-6: Advanced, requires solid domain expertise
- 7-8: Expert level, requires deep specialized knowledge
- 9-10: Cutting-edge, requires highly specialized or research-level expertise

Please respond with ONLY a single number (1-10) representing the difficulty score.

## D.2   REALISM EVALUATION PROMPT

We employ the following system prompt to evaluate the realism and feasibility of instructions:

**System Prompt:** You are an expert evaluator for assessing the realism and feasibility of instructions/prompts. Please evaluate the given instruction on a scale of 1-10 based on:

- Feasibility: Can this task actually be completed in the real world?
- Practicality: Does this request make sense in real-world scenarios?
- Realistic context: Are the assumptions and requirements reasonable?
- Real-world applicability: Would someone actually need this in practice?
- Logical consistency: Are there any contradictions or impossible requirements?

Scoring guidelines:

- 1-2: Completely unrealistic, impossible to execute, or nonsensical
- 3-4: Mostly unrealistic, major feasibility issues or impractical requirements
- 5-6: Somewhat realistic but has questionable assumptions or limited applicability
- 7-8: Quite realistic and feasible, minor issues but generally practical
- 9-10: Highly realistic, completely feasible, and practically useful

> Please respond with ONLY a single number (1-10) representing the realism score.

### D.3 QUALITY EVALUATION PROMPT

We employ the following system prompt to evaluate the quality of instruction-response pairs:

> **System Prompt:** You are an expert evaluator for assessing the quality of instruction-response pairs. Please evaluate the given instruction-response pair on a scale of 1-10 based on:
>
> - Accuracy and correctness of the response
> - Completeness and thoroughness of the answer
> - Clarity and coherence of explanation
> - Alignment between instruction and response
> - Educational value and helpfulness
> - Reflection of human preferences and real-world applicability
>
> Scoring guidelines:
>
> - 1-2: Poor quality, incorrect, unhelpful, or completely misaligned
> - 3-4: Below average, partially correct but lacks clarity or completeness
> - 5-6: Average quality, generally correct but could be improved
> - 7-8: Good quality, accurate, clear, and helpful
> - 9-10: Excellent quality, comprehensive, insightful, and highly valuable
>
> Please respond with ONLY a single number (1-10) representing the quality score.

## E HUMAN PREFERENCE EVALUATION TEMPLATE

We employ the following template to evaluate human preferences between model responses:

> **System Prompt:**
> You are a highly efficient assistant who evaluates and selects the best large language model (LLM) based on the quality of their responses to a given instruction. This process will be used to create a leaderboard reflecting the most accurate and human-preferred answers.
>
> **User Prompt:**
> I require a leaderboard for various large language models. I'll provide you with prompts given to these models and their corresponding outputs. Your task is to assess these responses and select the model that produces the best output from a human perspective.
>
> **Instruction:**
>
> ```
> {
>     "instruction": """{instruction}"""
> }
> ```
>
> **Model Outputs:**
> Here are the unordered outputs from the models. Each output is associated with a specific model, identified by a unique model identifier.
>
> ```
> {
>     {
>         "model_identifier": "m",
>         "output": """{output_1}"""
> ```

```
        },
        {
            "model_identifier": "M",
            "output": """{output_2}"""
        }
}
```

**Task:**
Evaluate the models based on the quality and relevance of their outputs, and select the model that generated the best output. Answer by providing the model identifier of the best model.
**Important:** We will use your output as the name of the best model, so make sure your output only contains one of the following model identifiers and nothing else (no quotes, no spaces, no new lines, etc.): m or M.

**Best Model Identifier:**

## F  CASE STUDY: COMPARATIVE ANALYSIS OF MODEL RESPONSES

To demonstrate our evaluation methodology, we present a comparative case study analyzing responses from three different models to an identical technical instruction. This comparison illustrates how different models approach complex technical tasks and provides insights into their relative strengths in meeting human preferences.

**Instruction:** "Provide a detailed guide on integrating GPRS technology into an embedded system, including selecting suitable modules, implementing a stable and efficient software stack, and ensuring reliable data transmission. The solution should be easy to use, require minimal technical expertise, and be compatible with other network protocols. Additionally, discuss best practices for optimizing performance and ensuring seamless integration with existing systems."

---

**GPT-4o Response**

**Integrating GPRS Technology into an Embedded System: A Detailed Guide**
Integrating **General Packet Radio Service (GPRS)** technology into embedded systems enables wireless data connectivity via cellular networks. This is particularly useful for IoT applications, remote monitoring, and telemetry systems. Below is a step-by-step guide covering **module selection, software integration, data reliability, compatibility**, and **performance optimization**.
**1. Selecting a Suitable GPRS Module**
Choosing the right GPRS module is crucial for ensuring reliable communication and seamless integration into the embedded system.
**Factors to Consider:**

- **Network Compatibility:** Ensure the module supports **quad-band (850/900/1800/1900 MHz)** for global connectivity.
- **Interface Support:** Common interfaces include **UART, SPI, I2C, or USB**.
- **Power Efficiency:** Look for power-saving features for battery-sensitive applications.
- **AT Command Support:** Modules leveraging **AT commands** allow easy interfacing with microcontrollers.
- **Additional Features:** Some modules support **TCP/IP stack**, **SMS**, or **GPS** capabilities.

**Recommended Modules**

1. **SIM800 Series (SIM800L, SIM800C, SIM800H)**
2. **SIM900**
3. **Quectel M95**
4. **Telit GE910**

**2. Hardware Integration**
To integrate the GPRS module into an embedded system, follow these hardware guidelines.
**Connecting the GPRS Module to a Microcontroller**

---

- **Power Supply:** Provide proper voltage levels (**usually 3.3V or 4.2V**) and sufficient current.
- **Serial Communication:** Use **UART TX/RX** to interface with the microcontroller for sending AT commands.
- **Antenna Connection:** Ensure proper RF antenna placement for signal reception.
- **SIM Card Slot:** Insert a valid SIM card with a data plan enabled.

**Example Circuit for SIM800L**

```
Microcontroller (ESP32/Arduino) → TX/RX → SIM800L (UART)
Microcontroller → GPIO → SIM800L Power Control
Microcontroller → GPIO → SIM800L Reset
VCC (3.7V-4.2V) → SIM800L
GND (Common Ground) → Microcontroller & SIM800L
```

[Content continues with detailed AT command examples, software stack implementation, reliability considerations, compatibility with network protocols, performance optimization, and integration best practices...]

---

**GPT-4o-mini Response**

Certainly! Here's a detailed guide on integrating GPRS technology into an embedded system, covering selection of modules, software stack implementation, reliable data transmission, ease of use, interoperability with other protocols, and best practices for performance and integration.

**Guide to Integrating GPRS Technology into Embedded Systems**

**1. Introduction to GPRS**

GPRS (General Packet Radio Service) is a packet-oriented mobile data standard on 2G and 3G cellular communication systems. It enables embedded devices to send/receive IP data over cellular networks, making it suitable for IoT, remote monitoring, and telemetry.

**2. Selecting Suitable GPRS Modules**

**Key Considerations:**

- **Compatibility:** Ensure the module supports standard AT commands.
- **Network Support:** Verify module supports the frequency bands of the operator in your region.
- **Data Speed:** Typical GPRS speeds (∼56-114 kbps) are sufficient for many sensor or telemetry applications.
- **Power Consumption:** Important for battery-operated devices; look for power-saving modes.
- **Interfaces:** UART, SPI, USB - choose based on your MCU's available interfaces.
- **Certifications:** FCC, CE, carrier approvals.
- **Size and Integration:** Choose a form factor that fits your hardware.

**Popular GPRS Modules:**

- SIM800, SIM900 (SIMCom) – widely used, affordable, robust, supports SMS, TCP/IP, FTP.
- Quectel M66 – compact, low power.
- Sierra Wireless HL Series – industrial grade with rich features.

[Content continues with hardware integration, software stack implementation using AT commands, reliability practices, protocol compatibility, ease of use considerations, performance optimization, integration guidelines, and a comprehensive summary checklist...]

Feel free to ask for code examples or hardware schematics if needed.

---

**o3-mini Response**

Below is a comprehensive guide for integrating GPRS technology into an embedded system. This guide is structured to cover hardware module selection, software stack implementation, reliable data transmission techniques, and best practices for optimizing performance and ensuring seamless integration with existing systems—all with an emphasis on ease of use and minimal technical overhead.

**1. Concept Overview**

GPRS (General Packet Radio Service) enables packet-switched data transfer over GSM networks. In an embedded system, GPRS adds wireless connectivity for transmitting small data packets (e.g., sensor data or remote control messages) without needing a full broadband connection. The integration process involves:

- Selecting a suitable GPRS module or modem
- Implementing a lightweight and robust software stack
- Managing reliable data transmission over variable networks
- Ensuring compatibility with other protocols (TCP/IP, MQTT, etc.)

---

**2. Selecting Suitable Hardware Modules**
**a. Evaluate Module Specifications**

- **Frequency Bands:** Make sure the chosen module supports the GSM frequency bands required by your target geographic region.
- **Data Speed:** For most sensor or control applications, modest speeds are fine; however, check that the module meets your throughput requirements.
- **Power Consumption:** Look for low-power modules if your design relies on battery power.
- **Operating Temperature & Environmental Specs:** Verify that the module can operate within your system's environmental constraints.

[Content continues with detailed module recommendations, hardware integration considerations, software stack implementation, reliability practices, performance optimization, compatibility guidelines, step-by-step integration process, and conclusion...]

This guide should provide the framework necessary to build a reliable, efficient, and user-friendly GPRS-enabled embedded system. Happy integrating!

**Content Structure and Presentation:** The three models demonstrate distinct approaches to content organization. GPT-4o employs a modern, visually-enhanced format with emoji integration and bold formatting that appeals to contemporary technical documentation standards. GPT-4o-mini follows a traditional academic structure with clear hierarchical organization and systematic progression. o3-mini adopts a professional documentation style with distinctive separator lines and methodical subsection organization.

**Technical Depth and Specificity:** GPT-4o provides the most specific technical details, including exact module part numbers, detailed AT command syntax with expected responses, and concrete implementation examples. GPT-4o-mini offers moderate technical depth with emphasis on general principles and decision-making criteria. o3-mini balances technical content with methodological considerations, focusing on software engineering best practices and systematic implementation approaches.

**Practical Applicability:** All responses address the instruction's requirement for minimal technical expertise, but through different strategies. GPT-4o achieves this through detailed step-by-step examples and ready-to-use code snippets. GPT-4o-mini emphasizes the use of existing libraries and provides decision-making frameworks. o3-mini focuses on modular design principles and abstraction layers that simplify long-term maintenance and development.

**Human Preference Alignment:** Based on typical user preferences for technical documentation, GPT-4o's response most closely aligns with modern expectations for actionable, immediately implementable guidance. Its combination of specific recommendations, working code examples, and visual appeal makes it highly practical for developers. GPT-4o-mini provides solid foundational knowledge suitable for users seeking to understand principles before implementation. o3-mini offers the most comprehensive long-term perspective, ideal for users planning scalable, maintainable systems.

# G   USE OF LARGE LANGUAGE MODELS IN PAPER WRITING

We used LLMs to assist with language polishing and minor formatting of the paper. No LLMs were involved in research ideation.

