# OpenReview forum: "WorldAlignment: Benchmarking Expert-Level Human Preference Alignment across Domains and Aspects"
_ICLR.cc/2026/Conference — Submitted to ICLR 2026_

### Official Review · Reviewer_R9CP · 2025-10-17

**Soundness:** 2
**Presentation:** 2
**Contribution:** 2
**Rating:** 2
**Confidence:** 4

**Summary:**

The paper introduces WorldAlignment, a new human preference alignment benchmark for evaluating large language models (LLMs). It aims to address the shortcomings of existing benchmarks like AlpacaEval 2.0 and WildBench, which focus on simple, instruction-following tasks and general user preferences. These benchmarks fail to assess complex, domain-specific, and nuanced scenarios adequately. In response, WorldAlignment provides a more robust, comprehensive evaluation by leveraging expert-level preference judgments across multiple domains, including instruction-following, mathematical reasoning, and code generation. The paper highlights the substantial performance gaps observed when evaluating state-of-the-art models, emphasizing the need for a benchmark tailored to real-world, expertise-driven tasks.

**Strengths:**

The paper provides a clear introduction to the limitations of existing benchmarks and effectively introduces WorldAlignment as a solution to these problems. The motivations for the proposed benchmark are well-articulated.

**Weaknesses:**

1. **Benchmark Comparison**: **Regarding coverage**, you mention that existing benchmarks primarily assess general instructions, which leads to the introduction of Instruction Following, Math, and Code in your work. However, Livebench also includes code and math subsets—how does your work innovate beyond that? Furthermore, why does the inclusion of code and math tasks justify calling your benchmark WorldAlignment? WildBench, which is constructed from real-world dialogues, also provides nuanced evaluation; however, you do not offer a Spearman correlation with ChatBot Arena for comparison. **Regarding difficulty**, you claim to introduce more challenging tasks, but from the prompt templates in Appendix C, it’s unclear why these prompts lead to more difficult questions. Your evaluation of "difficulty" using GPT-4o as a Judge is also somewhat limited—aren’t benchmarks like AIME, which are constructed manually, already difficult enough? **Regarding response length**, the analysis in Figure 2 suggests that longer instructions and responses correlate with higher diversity, yet it doesn’t clearly explain why longer prompts necessarily indicate greater difficulty. Do difficult tasks require long instructions? This reasoning is not entirely convincing. Furthermore, in terms of **bias control**, Section 3.1 and [2] show significant overlap—mainly just the addition of "d." Equations (2) and (3) also appear largely unchanged from [2], with no clear explanation of how "d" is reflected on the right-hand side, raising concerns about the novelty.

2. **Benchmark Construction Details**: There is a lack of detailed information regarding the construction of your benchmark. For example, how were the multiple domains under Instruction Following structured? What does "persona" refer to, and how is it defined? How do you ensure diversity and difficulty in the tasks? It seems that a single prompt is insufficient to achieve this. Appendix C doesn’t clarify these aspects, and the main text doesn’t provide further insight into how these benchmarks were constructed or how they ensure the claimed diversity and difficulty.

3. **Model Evaluation Bias**: The use of GPT-4o both as a reference model and as a judge introduces potential self-judging bias, as noted in [1]. To improve the robustness of the evaluation, it would be valuable to address this bias through alternative models or by incorporating more explicit control measures.

4. **Mathematical and Code Tasks**: The paper claims that post-trained models perform poorly on mathematical reasoning and code generation tasks. However, the open source models tested in Table 1 might not represent the state-of-the-art in these domains. Models like Qwen3 or DeepSeek-R1, which may have been specifically trained on mathematical and coding tasks, would provide a more appropriate baseline. Moreover, instead of relying on LLMs as judges, it would be useful to construct objective, standard answers for these tasks to ensure more consistent and accurate evaluation.

5. **Overclaim in Figure 1**: The paper suggests that traditional benchmarks primarily focus on Instruction Following. However, both Livebench and WildBench also include code and math tasks, which are not adequately acknowledged in your claims. Additionally, WildBench is constructed from real-world dialogues, not just Instruction Following. Therefore, the argument that the conventional scope is limited to Instruction Following is not entirely accurate.

**Questions:**

See above.

---

### Official Review · Reviewer_PWx4 · 2025-10-29

**Soundness:** 1
**Presentation:** 2
**Contribution:** 1
**Rating:** 2
**Confidence:** 4

**Summary:**

This paper proposes WorldAlignment, a novel benchmark for assessing LLM alignment. It critiques existing benchmarks like AlpacaEval 2.0 for their focus on simplistic instruction-following tasks and introduces a multi-domain dataset covering instruction-following, mathematical reasoning, and code generation. The benchmark comprises 2400 high-quality synthetic prompt-response pairs (800 per domain), generated using GPT-4o conditioned on diverse personas to ensure complexity and diversity.

**Strengths:**

1. The regression-based length correction and domain-aware modeling may provide a fairer, more robust metric than raw win rates.
2. Persona-guided synthetic generation yields diverse, challenging prompts (mean difficulty 7.21 vs. AlpacaEval's 3.20), assessed via feasibility and quality scores, supporting cost-effective benchmark updates.

**Weaknesses:**

1. Different from the authors' main claim that WorldAlignment is "an expert-level, multi-domain human preference benchmark" contradicts with its reliance on GPT-4o-generated pairs. It risks contamination and may not capture authentic human preferences, as real-world interactions (e.g., from Chatbot Arena or human annotators) could introduce.
2. The domains in WorldAlignment, instruction following has been extensively studied in AlpacaEval and ArenaHard. Coding and math have their corresponding objective benchmark like LiveCodeBench and AIME, MATH. As there are ground truths in code generation and math reasoning, it is not necessary to propose a synthetic benchmark for coding and math reasoning.
3. Using GPT-4o and GPT-4.1-Mini as judges introduces potential self-bias, with inconsistent ratings across domains, lacking human validation for gold-standard reliability.

**Questions:**

- In FIgure 2, what model is used for response generation? Why is positive correlation between instruction and response length better than weak relationship?
- The results that GPT-4.1 is better than GPT-5 on code generation is very wired, and conflicts with other coding benchmarks. Can you explain why?
- Is the proposed multi-domain regression model better than the original one? Do you have any experiment results?

---

### Official Review · Reviewer_sMQN · 2025-10-31

**Soundness:** 2
**Presentation:** 3
**Contribution:** 3
**Rating:** 4
**Confidence:** 4

**Summary:**

This paper introduces WorldAlignment, a multi-aspect human preference benchmark. It features complex preference pairs and incorporates domain-specific knowledge, providing a more rigorous standard for evaluation. This benchmark enables a more fine-grained assessment of current open-source and closed-source models.

**Strengths:**

1. This work introduces WorldAlignment, a benchmark that offers a more comprehensive evaluation of human preference alignment across multiple aspects and domains.

2. Additionally, this work proposes specialized evaluation metrics to better leverage the assessment capabilities of the WorldAlignment benchmark.

**Weaknesses:**

1. The data cleaning methodology is unspecified. It is unclear whether samples from various vertical domains were manually inspected by dedicated domain experts.

2. Employing GPT-4o to simultaneously generate baseline responses and act as the judge model may introduce unfairness or bias into the evaluation, as models tend to favor their own outputs.

3. The analysis of post-training effects is overly limited in its scope, encompassing a narrow range of model sizes and preference alignment methods. Consequently, the study fails to yield systematic or insightful conclusions. Further exploration is required across a wider variety of models (such as the Qwen series), a more diverse set of model sizes, and a broader spectrum of preference alignment or RL-based approaches. Additionally, the choice of training data itself may have a non-negligible influence on the results, a factor that warrants careful consideration.

**Questions:**

In Sections 3.1 and 3.2, the notation 'y' is used to represent both the preference label and the output from the model. This dual usage could be confusing for readers. I suggest using distinct notations to differentiate between them for improved clarity. For instance, you could use 'z' to represent the response in Section 3.2, which is the same as 3.1.

---

### Official Review · Reviewer_ySE5 · 2025-10-31

**Soundness:** 2
**Presentation:** 3
**Contribution:** 2
**Rating:** 2
**Confidence:** 4

**Summary:**

This paper presents WorldAlignment, a benchmark just like Alpaca Eval and Arena Hard to evaluate the model's general instruction following ability. The benchmark gathers responses sampled from GPT-4o as the baseline and uses a dual judge-system of GPT-4o and GPT-4.1-mini to evaluate the win rate of a "candidate model". The benchmark construction and eval protocol process is very similar to AlpacaEval and ArenaHard but the authors demonstrates that the prompts are more diverse and of higher quality. The author benchmarks many proprietary models and conducts evaluation by category (general instruction following, math, coding)

**Strengths:**

1. The paper studies an important problem. The current instruction following (IF) benchmarks are of low quality (e.g. Arena Hard has many weird prompts) and they have been extensively used now. We really really do need newer benchmarks for general chatbot evaluation.

2. The paper made efforts in controlling for the quality and diversity, and conducted analysis to make sure that their benchmark is indeed more difficult and higher quality.

3. The presentation of the paper is clear, with comprehensive results on both proprietary models and post-trained open-sourced models.

**Weaknesses:**

1. The issue (which is also an issue in Alpaca Eval and Arena Hard) is that they mixed Math and Coding prompts with general instruction following tasks, and adopts a fixed way (LM-judge) for evaluation. Math and Coding questions are tasks that the model either is or is not correct. For example, if you are asking a model to solve a math problem, then the evaluation should be whether the answer matches the groundtruth. If the question is about coding, then you need to pass unit tests. Adopting an LM-judge for all of such question doesn't seem correct to me. It would be much much better if the authors can focus on chatbot general instruction following tasks instead of trying to make it "multi-domain".

2. The prompts that the author used was sampled from GPT-4o. While it enhances the difficulty of the prompts, it results in a mismatch between how people generally use such models. There are natural prompts datasets (e.g. WildChat) out there and you can filter from it to control for quality. Manually conditioning on personas and sampling prompts from GPT-4o raises the question on how practical  / how aligned this eval procedure is to human judgement.

3. The author mentions that past works "are often susceptible to spurious correlations, including response length bias, formatting preferences, and positional effects" in the introduction. But I don't really see how the authors modify their benchmark design choices in resolving these problems. I think such problems are systematic and no matter how you control for the data quality, you will still have positional biases, and stylistic biases in LM-judges. For example, GPT-4o might favor a certain style of response that humans dislike.

**Questions:**

027 SoTa and post-trained models. I believe that the two are not complimentary? Maybe just use "frontier models"

191 - 192: As shown in Figure 2a, WorldAlignment instructions exhibit a substantially broader length distribution, extending to longer and more complex prompts.

Longer prompts/generations doesn’t necessarily mean that they are more complex. I wouldn't make this claim here.

---

### Meta-Review · Area_Chair_ECuo · 2025-12-03

**Summary:**

This paper introduces a new human-preference alignment benchmarks updated for the current generation of LLMs.  The reviewers raised several large criticisms: (1) code and math don’t require an LLM judge, (2) GPT4o prompts may be poorly reflective of human prompts, (3) not clear how benchmark adjusted to account for problems they list from previous benchmarks, (4) unclear data cleaning, (5) using GPT4o both to generate prompts and also as a judge may be problematic

**Reviewer Concerns:**

No rebuttals were posted, so no concerns were addressed.

**Reviewer Scores:**

The reviewer scores were 2, 4, 2, 2 with the authors not rebutting the reviews at all.  Therefore, I must assume that reviewers would all maintain their scores.

---

### Decision · Program_Chairs · 2026-01-26

Reject